# Host Immune Responses to *Clostridioides difficile* Infection and Potential Novel Therapeutic Approaches

**DOI:** 10.3390/tropicalmed8120506

**Published:** 2023-11-23

**Authors:** Md Zahidul Alam, John E. Markantonis, John T. Fallon

**Affiliations:** Department of Pathology and Laboratory Medicine, Brody School of Medicine, East Carolina University, 600 Moye Boulevard, Greenville, NC 27834, USA; markantonisj22@ecu.edu (J.E.M.); fallonj19@ecu.edu (J.T.F.)

**Keywords:** *Clostridioides difficile* infection, microbiome-based therapy, mucosal immunity, recurrent *C. difficile* infection, innate lymphoid cells

## Abstract

*Clostridioides difficile* infection (CDI) is a leading nosocomial infection, posing a substantial public health challenge within the United States and globally. CDI typically occurs in hospitalized elderly patients who have been administered antibiotics; however, there has been a rise in the occurrence of CDI in the community among young adults who have not been exposed to antibiotics. *C. difficile* releases toxins, which damage large intestinal epithelium, leading to toxic megacolon, sepsis, and even death. Unfortunately, existing antibiotic therapies do not always prevent these consequences, with up to one-third of treated patients experiencing a recurrence of the infection. Host factors play a crucial role in the pathogenesis of CDI, and accumulating evidence shows that modulation of host immune responses may potentially alter the disease outcome. In this review, we provide an overview of our current knowledge regarding the role of innate and adaptive immune responses on CDI outcomes. Moreover, we present a summary of non-antibiotic microbiome-based therapies that can effectively influence host immune responses, along with immunization strategies that are intended to tackle both the treatment and prevention of CDI.

## 1. Introduction

### 1.1. Overview of Clostridioides difficile Infection (CDI)

The anaerobic gram-positive bacterium *Clostridioides difficile*, commonly referred to as *C. diff*, is one of the leading causes of antibiotic-associated diarrhea and infectious colitis in the world [1,2]. The organism commonly colonizes the gastrointestinal tracts of both healthy and chronically ill individuals of all age groups [3,4,5]. Disruption of the normal gut microbiota through administration of antibiotics can result in the proliferation of *C. difficile* with toxin production in capable strains resulting in CDI [4,6,7]. Most toxigenic strains of *C. difficile* produce toxin B (cytotoxin), with or without toxin A (enterotoxin) [7,8,9]. Some hypervirulent strains, such as Ribotype 027, produce a binary toxin (CDT), which rearranges the actin cytoskeleton of enterocytes, detrimentally affecting multiple cellular processes [1,7,8].

Accurate and timely diagnosis of CDI is an important yet challenging issue facing healthcare systems today [3,4]. CDI is a common misdiagnosis for many patients, due to the high frequency of *C. difficile* colonization, varying individual tolerances to *C. difficile* toxins, numerous causes of diarrhea besides *C. difficile*, and common pitfalls of routine diagnostic testing [3,4,5,6,10]. Prompt initiation of appropriate treatment is needed to prevent the development of severe manifestations of CDI (e.g., pseudomembranous colitis, toxic megacolon, ileus, septic shock); however, unnecessary treatment can lead to the establishment of multidrug-resistant organisms in patients’ colonic flora [3,4,5,6]. Diagnostic stewardship, utilizing astute clinical judgment and judicious use of microbiological testing, is key to maximizing diagnostic accuracy [3,4,5,6].

Given the challenges associated with accurately diagnosing CDI, recent advancements in clinical tests and therapies have become increasingly significant in improving patient outcomes. According to a review of the diagnosis and management of CDI in adults, new clinical tests and therapies have become available, and clinical practice guidelines were updated [11]. The Infectious Diseases Society of America (IDSA) and the Society for Healthcare Epidemiology of America (SHEA) have published a clinical practice guideline on the management of CDI in adults [12]. A systematic review of guidelines for the diagnosis and treatment of CDI was also published [13].

### 1.2. Public Health Significance of CDI

With just over 200,000 cases annually resulting in ~12,000 deaths, CDI is a major public health concern in the U.S [14]. The Centers for Disease Control and Prevention (CDC) lists *C. difficile* as an urgent threat requiring aggressive preventative measures [14]. Reducing the amounts of CDI cases by optimizing modifiable risk factors (e.g., unnecessary antibiotic exposure, hospitalization prolongation) not only reduces the associated morbidity and mortality of CDI but also leads to a substantial reduction in the substantial healthcare costs associated with this infection [4]. It is estimated that CDI costs USD 1 billion in associated U.S. healthcare spending, based on the most recent CDC statistics [14].

Stringent infection prevention and control practices are required to prevent the spread of this spore-forming bacterium in healthcare settings [3,4]. Recommended infection control practices include isolation of patients with CDI, wearing gloves and gowns with disposable equipment use during patient encounters, hand washing with soap and water after patient contact, and use of sporicidal cleaning agents [3,4]. Antibiotic stewardship programs are effective in reducing CDI cases and should be established in most healthcare systems [3,4]. Reducing unnecessary antibiotic administration and utilizing antimicrobial agents with lower association with CDI should be the goal of these programs [4].

### 1.3. The Challenge of CDI Recurrence

One of the most challenging issues facing the management of CDI is its propensity for recurrence, commonly resulting in multiple subsequent episodes [3,12,15]. Not only does this contribute to this disease process’s morbidity and mortality, but it also leads to an exponential rise in healthcare-associated costs [12,16]. Disruption of the gut microbiota leading to the development of CDI and following treatment of it predisposes individuals to recurrence [17]. The inability of the host’s immune system to fully clear the organism from the gastrointestinal tract is also a major factor in recurrence [17].

Several new treatment options have shown promise in reducing the frequency of CDI recurrence [3,12,18]. Vancomycin and metronidazole have been the two agents most commonly used for the treatment of CDI; however, both demonstrate antibacterial effects against beneficial bacteria in the normal gut flora, leading to persistent dysbiosis [3,12,19]. Fidaxomicin has become the preferred treatment agent in non-fulminant CDI cases, due to its narrow-spectrum antibacterial activity leading to the preservation of the gut microbiota and less CDI recurrences [3,12,19,20]. The MODIFY I and MODIFY II clinical trials showed that bezlotoxumab, a monoclonal antibody targeting the *C. difficile* toxin B, is effective in reducing recurrence following primary and recurrent CDI when administered with standard-of-care antibiotics [21]. Bezlotoxumab administration should be considered in patients with recurrence or at high-risk for recurrence [3,12,19]. Fecal microbiota transplantation (FMT) is a novel option for individuals with multiple CDI recurrences despite appropriate antibiotic therapy [3,12,19,22,23]. Stool from donors with healthy gut microbiota are transplanted via colonoscopy or oral capsule to restore a functional gut flora [24]. FMT has been shown to be highly successful in preventing CDI recurrence, but the transplantation of multi-drug resistant organisms and toxin-producing bacteria have been reported [22,23,25,26,27,28]. Although antibiotics are the primary treatment for initial CDI, their effectiveness is only partial. Additionally, the use of antibiotics can lead to persistent dysbiosis, contributing to recurrent infections in a significant number of patients [15,29,30]. Recent studies highlight the role of immune responses and cytokines released during acute CDI in the disease’s pathogenesis. Specifically, the modulation of cytokines has been proven to influence CDI outcomes. For instance, elevating the levels of certain cytokines, such as IL33 and IL25, enhances host protection [31,32]. Mice treated with these cytokines exhibited less mucosal damage and greater resistance against CDI compared to the untreated group. Furthermore, depleting immune cells has been shown to impact the outcome of CDI pathogenesis [33,34]. These findings underscore the potential of host immune-based therapy in managing CDI. However, our understanding of the host immune response during CDI remains incomplete. In the next section, we delve into the immune responses mounted against CDI.

## 2. Host Immune Response to *C. difficile* Infection

### 2.1. Innate Immune Response

The innate defense mechanisms against *C. difficile* infection include the endogenous microbial flora, the mucus barrier, intestinal epithelial cells, and the mucosal immune cells. In this review we limit our discussion to mucosal immune response and the associated immune cells.

*C. difficile* primarily produces toxin A and toxin B (with certain strains also producing binary toxins), causing disruption to the intestinal epithelium, leading to activation of the immune responses in the lamina propria of the colon. These toxins have a profound impact on innate immune defenses, triggering the release of various proinflammatory mediators, such as cytokines and chemokines, promoting the recruitment and activation of diverse innate immune cells (Figure 1). Further, the disrupted mucosal barrier, due to toxin-mediated damage, allows commensal bacteria to translocate into the lamina propria and systemic circulation, leading to robust inflammatory responses.

In response to *C. difficile* toxins and their associated damage, intestinal epithelial cells, and innate immune cells in the lamina propria release proinflammatory cytokines (IFNγ, IL-12, IL-6, IL-23, IL-1β, etc.) and chemokines (CXCL1, CXCL2, and CXCL5), leading to the recruitment of neutrophils to the site of infection [35,36,37,38].

Proteins involved in the signaling of the innate immune response, such as nucleotide-binding and oligomerization domain 1 (NOD1), myeloid differentiation factor 88 (MyD88), and adaptor protein for inflammasome, known as apoptosis-associated speck-like protein containing a CARD (ASC), play a role in *C. difficile* pathogenesis. Studies show that mice deficient in Nod1, MyD88, and ASC signaling have decreased levels of CXCL1 and infiltrating neutrophils in their colons, which is associated with enhanced disease severity and mortality, compared to wild-type mice [33,39,40]. It is worth mentioning that the role of neutrophils and their effect on host factors in CDI susceptibility is complex and context-dependent, with various pathways and cell types involved in the overall pathogenesis. For example, Jarchum et al. found that depletion of neutrophils in mice through antibody-mediated Gr1^+^ (Ly6G) depletion increases the disease severity and mortality during CDI [33]. This finding is supported by the clinical observation that in hospitalized leukemia and allogeneic hematopoietic stem cell transplant patients, neutropenia is positively correlated with CDI susceptibility and recurrent CDI [41,42]. In contrast, another study found that blocking of neutrophil infiltration in the colon by anti-CD18 (leukocyte adhesion molecule) treatment in rabbit leads to decreased Toxin A-mediated enterotoxicity, compared to non-anti-CD18 treated mice [43]. Similarly, another study in a mouse model of CDI showed that mice treated with macrophage migration inhibitory factor (MIF) blocking antibody demonstrated improved disease severity and survival, which is associated with reduced neutrophil recruitment in the colon [44]. However, a recent study by Chen et al. found no differences in CDI severity when they depleted neutrophils using anti-Ly6G antibodies [34]. The differing roles of neutrophils in CDI outcomes in various studies may result from variances in animal models, *C. difficile* strains, experimental design, and host-specific factors. Other innate immune cells, such as eosinophils, play a protective role in CDI. In studies using mice, Buonomo et al. show that mice adoptive transferred of eosinophils are more protected against CDI [31], supported by a clinical study showing that, in patients, the eosinopenia in peripheral blood at the time of CDI diagnosis was associated with higher mortality [45].

*C. difficile* and its toxins can activate surface and intracellular innate immune sensors, including TLR4, TLR5, and inflammasome signaling pathways [39,46,47,48]. Depletion of TLR4-signaling pathways in mice results in an increased bacterial burden and disease severity during CDI [46]. On the other hand, TLR5 deficiency did not affect survival, but stimulating TLR5 with flagellin provided protection against CDI, likely due to its positive impact on the intestinal epithelial layer [47].

In vitro, infection of murine peritoneal macrophages with a toxigenic strain of *C. difficile* results in the release of the proinflammatory cytokine pro-IL-1β, a process that is dependent on MyD88 and, to some extent, TLR2 [49]. In the same study, toxigenic *C. difficile* activated inflammasome through the ATP-P2X_7_ pathway, leading to caspase-1-dependent pyroptosis [49]. In a different study, stimulating the J774A.1 murine macrophage cell line with *C. difficile*’s surface layer protein (SLP) led to increased expression of TLR2, TLR4, and MHCII [50]. Furthermore, the activation of TLR4 using SLPs triggered p38 signaling, leading to increased production of IL-1β, IL-6, TNF-α, IL-12p40, and the upregulation of chemokines like MIP-1α, MIP-2, and MCP [50].

Recent studies have shed light on the protective role of innate lymphoid cells (ILCs) in acute CDI. Unlike adaptive immune cells like T and B cells, ILCs, which include ILC1, ILC2, and ILC3, lack antigen-specific receptors [51]. However, they possess the ability to detect cytokines and chemokines and are pivotal in orchestrating immune responses against infections and maintaining tissue homeostasis [51]. During CDI, when commensal microorganisms, pathobiont bacteria, or *C. difficile* toxins translocate across the gut barrier into deeper tissues, this triggers the activation of ILCs. A study showed that Nfil3-deficient (Nfil3^-/-^) mice, which exhibit defects in the development and functionality of ILCs, demonstrated heightened susceptibility to acute CDI compared to their wild-type counterparts [52]. Nfil3^-/-^ mice infected with *C. difficile* manifest increased weight loss and higher mortality rates than their wild-type (WT) counterparts. Another study conducted by Abt and colleagues uncovered the protective function of IFNγ-expressing ILC1 as part of the host’s defense against CDI, complemented by the role of IL22-expressing ILC3 [53]. Additionally, another research group has demonstrated the role of ILC2s in protection against *C. difficile* colitis. For instance, Frisbee et al. showed that ILC2 provides defense against acute CDI, especially when activated by IL-33 [32]. Notably, treating *C. difficile*-infected mice with IL-33 leads to reduced neutrophil levels and increased eosinophils in the colon, resulting in type 2-associated mucosal immunity. Similarly, another study further underscored the IL-25-mediated role of ILC2s in defending against CDI [31]. In parallel, other researchers also demonstrated the protective role of ILC3 in safeguarding against CDI, primarily through the interleukin-22-mediated effect [54,55].

The aforementioned studies indicated the crucial role of innate immunity in the initial containment of infections and its contribution to resolving CDI. Nonetheless, the recurrence of episodes implies that adaptive immune cells may be fundamentally involved. This is attributed to their capacity to respond in an antigen-specific manner and generate memory, thereby providing additional protection.

### 2.2. Adaptive Immune Response

The defining feature of the adaptive immune system is the clonal expansion of lymphocytes, resulting in a durable and highly specific response. Typically, memory T and B cells orchestrate more robust and rapid immune reactions to pathogens—rapidly multiplying, generating effector cytokines, and executing various effector functions [56]. However, an impaired immune response to pathogens can result in recurring infections.

Around 20 to 35% of individuals treated for CDI encounter at least one additional episode within 2 to 8 weeks of their initial CDI treatment [18,42]. These subsequent events can manifest as either a relapse with the same *C. difficile* strain or as a reinfection with a different strain [57]. One possible factor contributing to susceptibility to recurrent CDI is the ongoing disturbance of gut microbiota diversity [15,58]. This, in conjunction with a weakened host response, may play a role, since low levels of serum antibodies against toxins A and B have been linked to the CDI recurrences [59,60].

In CDI, immunoglobulins such as systemic IgG and mucosal IgA significantly influence disease outcomes. In human CDI cases, the disease’s severity shows an inverse correlation with the levels of toxin-specific IgA and IgG antibodies in the serum and secretory intestinal IgA [60,61]. Furthermore, in fecal samples, individuals experiencing a single CDI episode demonstrated significantly higher anti-TcdA IgA titers, compared to those with recurrent CDI [62]. Studies also have found an inverse relationship between IgG levels against toxin B and disease severity, as well as CDI recurrence. Notably, patients possessing antibodies against toxin B exhibit enhanced protection compared to those with antibodies against toxin A [63]. Further, monoclonal antibodies targeting TcdB have demonstrated greater efficacy in CDI treatment [21], indicating TcdB’s higher toxicity compared to TcdA. It is important to note that while elevated antibodies against *C. difficile* toxins can guard against the disease and its relapse, they do not prevent *C. difficile* colonization in the colon.

Most of the research pertaining to the host’s adaptive immunity in CDI has primarily concentrated on humoral responses. On the other hand, the importance of T follicular helper (Tfh) cells, which play a critical role in generating plasma cells that produce antibodies and long-lasting memory B cells, has only recently begun to be investigated. The exploration of various subsets of T cells and their effector functions against *C. difficile* is still in its early stages.

At the germinal center, Tfh cells play a crucial role in aiding B cells, ultimately leading to the differentiation of activated B cells into plasma cells and memory B cells [64,65]. Consequently, Tfh cells are instrumental in conferring antibody-mediated protection to the intestinal mucosa against pathogens. In a murine model of recurrent *C. difficile* infection, Amani et al. observed a notable expansion in the population of lymph node resident Tfh cells, encompassing both germinal center and non-germinal center Tfh cells, following CDI when compared to uninfected mice [66]. Despite the observed expansion of Tfh cells, the B cell response proved insufficient in preventing disease recurrence. Furthermore, the authors demonstrated that CDI failed to elicit a robust B cell memory response.

Studies have demonstrated that *C. difficile* strains trigger a CD4^+^ T cell response. For example, the hypervirulent *C. difficile* R20291 strain elicits a robust Th1 and Th17 response, as evidenced by an increased presence of IFNγ^+^ and IL-17A^+^ CD4^+^ T cells, when compared to the non-virulent *C. difficile* 630 strain in co-culture with murine splenocyte and bone-marrow-derived dendritic cells [67]. In a clinical study, *C. difficile*-infected patients exhibited a shift in immune responses from Th1 to Th17 or Th2 as disease severity increased [68]. Studies by other investigators also demonstrated the role of T cells in CDI. Notably, young children display resistance to CDI, and a study in mice has highlighted the role of IL-17A produced by γδ T cells in this resistance [34]. Neonatal mice, resistant to CDI, demonstrated substantial IL-17 production by RORγt^+^ γδ T cells [34]. However, this protective effect was lost upon depletion of these IL-17-producing T cells.

Regulatory T cells (T_reg_), a subset of CD4^+^ T cells, are crucial for maintaining intestinal immune tolerance and homeostasis. Their specific impact on a host’s susceptibility to acute CDI and relapse remains unclear. However, a recent study highlighted the vital role of T_reg_ cells in successfully engrafting fecal microbiota and clearing chronic CDI in mice [69]. Mechanistically, depleting T_reg_ cells causes an exaggerated immune response in the colon, hindering *C. difficile* clearance by impeding the engraftment of microbial populations derived from fecal microbiota transplantation (FMT).

A deeper understanding of the adaptive immune response is crucial for advancing our knowledge of recurrent infection pathogenesis and the development of *C. difficile* vaccines. While the innate immune response has received significant attention, the adaptive immune response remains understudied. To effectively combat *C. difficile*, we must delve into comprehensive investigations of Th cell responses, including detailed characterizations of their phenotypes and functions. This exploration is essential to assess how these populations influence various stages of CDI and their contributions to the disease’s pathogenesis or resolution. The fundamental question of whether adaptive immunity affects the clearance or persistence of *C. difficile* remains to be definitively answered.

## 3. Microbiome-Based Treatment Approach to Treat CDI

The current treatment approach for CDI primarily relies on antibiotics, such as metronidazole, vancomycin, and fidaxomicin [3,13]. However, a significant downside of antibiotics, particularly metronidazole and vancomycin, is their impact on the beneficial indigenous flora [29,30]. Fidaxomicin offers a promising alternative, with a lower *C. difficile* recurrence rate [70,71]. Fidaxomicin’s narrower antimicrobial spectrum results in less dysbiosis [72]. Antibiotic-induced persistent dysbiosis contributes to CDI recurrence. Therefore, recent research has explored alternative treatments that aim to prevent CDI recurrences by enhancing gut bacteria and metabolites, as well as influencing host immune responses to improve CDI outcomes.

### 3.1. Probiotics

Probiotics have shown promise in preventing the relapse of CDI in patients. Probiotics are live microorganisms, such as bifidobacterium, saccharomyces, and lactobacillus, that, although not naturally occurring in the host, offer health benefits upon administration [73]. Studies have uncovered their anti-inflammatory role in improving colitis in mice and have proven beneficial in the treatment of individuals with ulcerative colitis [74,75,76]. In the context of CDI, *Saccharomyces boulardii*, when administered alongside standard antibiotic therapy, has demonstrated potential in preventing recurrence [77]. For instance, one study by McFarland and colleagues reported that four weeks of oral *S. boulardii* supplementation in conjunction with antibiotics significantly reduced recurrent *C. difficile*-associated disease (34.6%), compared to a placebo group (64.7%) [77]. Similarly, another study [78] found *S. boulardii* to be effective in decreasing recurrences (16.7%) when used with a high dose of vancomycin (2 g/day), compared to a placebo combined with a high-dose vancomycin regimen (50%). In both studies, the effectiveness of *S. boulardii* was observed when administered alongside antibiotic therapy. Nevertheless, differing outcomes were observed in other studies. For example, a multicenter, randomized, double-blind, placebo-controlled study conducted by Allen et al. found no benefit of probiotics in preventing CDI [79]. The study focused on inpatients aged 65 years and older, exposed to one or more oral or parenteral antibiotics. Participants received either a multistrain preparation of lactobacilli and bifidobacteria for 3 weeks or an identical placebo. The primary outcomes were the occurrence of antibiotic-associated diarrhea (AAD) within 8 weeks and *C. difficile*-associated diarrhea (CDD) within 12 weeks of recruitment. A total of 1493 participants were randomly assigned to the lactobacilli and bifidobacteria group, and 1488 to the placebo group. The study concluded that there was no evidence supporting the effectiveness of a multistrain preparation of lactobacilli and bifidobacteria in preventing AAD or CDD. Another study, conducted by Heil et al., yielded similar outcomes [80]. That investigation aimed to evaluate the impact of a computerized clinical decision support (CCDS) tool on prescribing probiotics for the primary prevention of CDI in adult hospitalized patients. Implemented across four hospitals, the study utilized electronic medical records to prompt probiotic prescriptions during antibiotic administration for high-risk patients. Unexpectedly, the post-intervention period witnessed an increase in CDI incidence, contrary to the anticipated benefits. The odds of CDI were 1.41 times higher in eligible patients post-intervention, signaling a lack of a protective effect from probiotics containing *Lactobacillus acidophilus*, *Lactobacillus casei*, or *Lactobacillus rhamnosus*. This study demonstrated that employing probiotics for the primary prevention of CDI in adult inpatients receiving antibiotics, guided by a CCDS tool, lacks substantial support. These studies underscored the imperative for a cautious reassessment of probiotic use in this specific clinical context. It is worth noting that probiotic therapy may have limitations, due to these bacteria’s inability to permanently colonize the disrupted intestinal environment and fully restore microbiome diversity.

### 3.2. Live Biotherapeutics

Live biotherapeutics offer promising treatment options for patients dealing with recurrent CDI. Unlike probiotics, these biotherapeutics consist of specific bacterial species or combinations designed to colonize the intestine, targeting particular diseases [81]. The intricate nature of microbial interactions with the host makes it challenging to pinpoint individual bacterial species or small combinations that can deliver therapeutic effects to recurrent CDI patients. Clinical trials have underscored the potential of this precision microbiome-based approach. In the realm of recurrent CDI treatment, a recent introduction is SER-109, an oral microbiome therapy comprised of Firmicutes spores administered after standard antibiotic treatment to combat recurrence [82]. Notably, the US Food and Drug Administration (FDA) has granted approval to Seres Therapeutics’ SER-109 for the prevention of recurrent CDI [83]. The FDA’s decision was based on the outcomes of the ECOSPOR III trial, which involved 182 participants with three or more CDI episodes within a year [83]. Another live biotherapeutic, VE303, consisting of a selected collection of eight bacteria developed by Vedanta Biosciences, demonstrated promise in phase 2 clinical trials aimed at treating recurrent CDI [84]. These studies demonstrated the emerging role of live biotherapeutics to potentially offer a safer and more effective treatment approach to treat CDI. However, the precise mechanisms through which live biotherapeutics offer protection against CDI are not yet fully understood. Nevertheless, it is believed that the restoration of a dysbiotic gut environment with beneficial bacteria and their associated metabolites is one potential mechanism [83,85,86].

### 3.3. Fecal Microbiota Transplantation (FMT)

Fecal microbiota transplantation (FMT), also known as bacteriotherapy, has garnered significant attention from the medical and scientific communities, due to its remarkable efficacy in treating recurrent CDI. Rigorous, double-blinded studies have established its effectiveness, achieving approximately an 89% success rate in preventing recurrences of this infection [18]. The FMT procedure involves the transfer of fecal material from a healthy donor to the patient’s intestinal tract, aiming to restore the recipient’s gastrointestinal bacterial diversity and the associated bacterial-derived metabolites, referred to as the metabolome [83,87]. The modern therapeutic use of FMT commenced in 1958, when Eisenmen and colleagues pioneered its application for treating pseudomembranous colitis [88]. Presently, FMT is primarily recognized for its effectiveness in managing recurrent CDI. On November 30, 2022, the FDA announced Rebyota as a preventive measure for the recurrence of CDI in individuals aged 18 and above who have undergone antibiotic treatment for recurrent CDI [89]. Rebyota is a pre-packaged, single-dose 150 mL microbiota suspension designed for rectal administration. It comprises a liquid mixture containing trillions of live microbes [89,90,91]. The microbiota suspension is prepared from stool donated by qualified individuals. The effectiveness of Rebyota is assessed through randomized, double-blind, placebo-controlled, multicenter studies [89]. Studies revealed that Rebyota is well-tolerated and safe for use in adults with recurrent CDI.

The precise mechanism underlying FMT’s efficacy remains incompletely elucidated, but it is thought to be multifaceted [85]. It involves the restoration of beneficial bacteria and metabolites (Figure 2), conferring resistance to *C. difficile*, while also influencing host immune responses that can impact the outcome of CDI [18,85].

During CDI, there is a notable reduction in the levels of SCFAs (short-chain fatty acids) and secondary bile acids within the colons of affected patients [92]. Following FMT, the observed clinical improvement in these patients is closely linked to the successful establishment of donor microbes, the restoration of SCFAs and secondary bile acids, and the subsequent reduction of inflammation in the colon [92,93]. The main factor contributing to FMT’s effectiveness in treating CDI is thought to be the restoration of colonization resistance against *C. difficile* by commensal bacteria, achieved by increasing the richness and diversity of gut flora and their associated metabolites. Supporting this widely accepted theory, it is important to note that SCFAs and secondary bile acid metabolites can directly impede the growth of *C. difficile* [84,86,94]. For example, commensal bacteria like *Clostridium scindens*, which produce secondary bile acids, play a role in directly inhibiting the growth of *C. difficile* [95].

Recent research in both human and animal studies revealed significant alterations in the host immune system subsequent to FMT, highlighting its therapeutic role. In a pilot study, successful FMT for recurrent CDI led to the upregulation of the bile acid-driven FXR-FGF signaling pathway in the ileum [96]. This resulted in increased fibroblast growth factor (FGF)-19 and decreased FGF-21 levels in the patient’s serum, and the pathway’s upregulation was associated with the restoration of the intestinal microbiome and secondary bile acid profile in the patient’s colon [96]. Another study by Marie and colleagues reported immunoregulatory changes following FMT, including increased levels of IL-25 in the colon [97]. IL-25 promotes type 2 immunity, which offers protection against acute CDI in mice by shifting the host’s response away from a pathogenic, proinflammatory state [7,31]. Furthermore, FMT was found to suppress proinflammatory immune responses, while enhancing the expression of a family of homeobox and laminin genes that support the development and homeostasis of the colon [97]. Additionally, Th17 cells in the peripheral blood decreased after FMT. In contrast, another study observed an increase in *C. difficile* toxin B-specific Th17 cells, as well as toxin A- and toxin B-specific IgG and IgA antibodies in the blood after microbial engraftment [98], suggesting that an augmented *C. difficile* toxin-specific adaptive immune response could be a key mechanism behind FMT’s efficacy.

Collectively, the observed alterations in the immune response subsequent to FMT support the idea that, beyond direct interactions between bacteria, FMT may enhance the host’s immune defenses against CDI. While animal and human trials involving FMT have illustrated its feasibility, they have also underscored the potential risks [25,26] associated with employing an unspecified bacterial consortium as a therapeutic approach and encouraged the exploration of more sophisticated alternatives, like a refined form of FMT. For instance, a purified mixture of isolated gut bacteria derived from healthy donors, referred to as ‘defined gut microbial ecosystem components,’ effectively eradicated *C. difficile* in patients who had previously failed to clear the pathogen through multiple rounds of conventional antibiotic therapy [99,100]. This defined FMT approach offers the promise of a safer, more regulated, and more widely accepted treatment method. Nonetheless, the efficacy of this approach needs to be confirmed through future large-scale randomized controlled trials.

## 4. Active and Passive Immunization Strategies against CDI

*C. difficile* releases toxin A and toxin B, which play a critical role in the pathogenesis of CDI. These toxins bind to cell receptors in intestinal epithelial cells, undergo endocytosis within the target cells, and, subsequently, cause glycosylation of host GTPases proteins [48,101]. This glycosylation disrupts cytoskeletal organization, leading to damage in the intestinal epithelium. The active immunization strategy targeting CDI involves activating the host’s immune system to generate antibodies in response to weakened toxins (toxoids), toxin fragments, or cell wall components of *C. difficile* introduced by the vaccine [102,103]. Most of the research into vaccine development against CDI is primarily centered on these toxins. Many efforts in developing effective vaccines against CDI is ongoing and showed promising results.

The Sanofi toxoid vaccine and the Valneva recombinant attenuated toxin vaccine have demonstrated the ability to induce robust antibody responses and offer substantial protection against CDI. Sanofi’s vaccines consist of formalin-inactivated toxin A and toxin B, purified from the VPI 10463 strain of *C. difficile*, and are combined with alum adjuvant [104]. The Valneva vaccine comprises a recombinant chimeric protein that encompasses the C-terminal binding domain of TcdA (15 of 31 repeats) and TcdB (23 of 24 repeats), connected with a sequence of 12 amino acids [105]. However, this vaccine presents certain limitations, as it lacks specific neutralizing epitopes found in the glucosyltransferase domain and binding regions of TcdB. Due to potential amino acid variations in the binding domain among different TcdB subtypes, this vaccine may not be effective against a range of clinical isolates. Pfizer developed a toxoid vaccine, PF-06425090, which elicited a strong antibody response [106,107]. This vaccine involved the genetic and chemical detoxification of TcdA and TcdB, with the addition of alum adjuvant. The clinical testing is ongoing for this vaccine.

Another vaccine, developed by Shire (NCT01259726), is currently undergoing clinical trials [108]. This Shire vaccine utilizes a live non-toxigenic strain of *C. difficile* administered through mucosal delivery. In addition, numerous studies have explored the use of a plasmid containing the receptor-binding domain of both TcdA and TcdB, which was tested in in vitro culture systems and animal models [109,110]. These studies demonstrated its ability to stimulate a B cell response and induce the formation of neutralizing antibodies.

In a study, researchers engineered a strain of *Lactococcus lactis* to produce recombinant fragments of TcdA and TcdB [111]. Mice orally vaccinated with this modified strain experienced reduced mortality and milder disease symptoms. The vaccinated mice exhibited higher levels of anti-TcdA/TcdB antibodies compared to the control group, and these antibodies effectively neutralized the toxins in vitro. Other studies also yielded promising results with bacterially delivered TcdA/TcdB vaccines. Hong and colleagues, for instance, engineered *Bacillus subtilis* spores to express the TcdA C-terminal. When mice received oral vaccination with the modified spores, they developed a robust IgA and IgG response against TcdA, which also exhibited cross-reactivity with TcdB [112]. This vaccine protected three-fourths of hamsters from mortality, and all surviving hamsters remained immune to re-challenge with *C. difficile*. This observation not only highlighted the efficacy of a vaccine delivery platform based on *B. subtilis* but also provided evidence that a TcdA antigen can elicit antibody responses with the capacity to neutralize both TcdA and TcdB.

In addition to studies on active immunization with *C. difficile* toxins, numerous investigations have explored alternative immunization strategies that can impede *C. difficile*’s colonization. One example is the surface-layer proteins (SLPs), which encompass the exterior of *C. difficile* bacteria and play roles in adhesion and immune activation. These SLPs primarily consist of high molecular weight SLP (HMW SLP) and low molecular weight SLP (LMW SLP), formed through SlpA cleavage by the protease Cwp84 [113]. Immunization with HMW SLP and LMW SLP has shown some promise in mice and hamsters, although SLPs are not highly immunogenic, prompting the exploration of adjuvant options for future studies [103]. Additionally, several spore core proteins, including CotA, CdeC, CotE, and CdeH, have been investigated as potential vaccine targets [114,115]. Flagellin FliC, a whip-like appendage that helps in *C. difficile*’s motility, has also been identified as a potent vaccine candidate [116,117,118].

Passive immunization is another method to mitigate CDI. Passive immunization involves delivering neutralizing antibodies directly into the body, offering several advantages such as high specificity and low toxicity of purified antibodies. This approach aims to provide patients with neutralizing anti-toxin antibodies to prevent recurrent *C. difficile* infection (rCDI). Notably, the FDA has approved bezlotoxumab, a monoclonal antibody targeting toxin B for the treatment of rCDI [20,119]. Bezlotoxumab has shown significantly better efficacy in preventing rCDI than actoxumab (antibody against toxin A) [21,120]. While bezlotoxumab has gained clinical approval for rCDI prevention, both bezlotoxumab and actoxumab have only been assessed as intravenous, systemic passive vaccine candidates [20,21]. These monoclonal antibodies are co-administered with antibiotic therapy. Future research should focus on evaluating these monoclonal antibodies as potential mucosal vaccines, particularly exploring the potential of bezlotoxumab as an independent preventive measure against rCDI.

## 5. Conclusions and Future Direction

In conclusion, the host immune response to *C. difficile* infection is a complex interplay of innate and adaptive mechanisms. The innate immune system is activated mainly by *C. difficile* toxins and its associated intestinal damage, leading to the release of proinflammatory mediators and the recruitment of various innate immune cells, including neutrophils and eosinophils. The role of neutrophils in CDI outcomes remains context-dependent, with differing results in various studies. Adaptive immunity, particularly humoral responses, plays a crucial role in protecting against CDI, as evidenced by the significance of toxin-specific antibodies in disease severity and recurrence. However, there is a need to prioritize research into T cell differentiation and activation mechanisms against *C. difficile*. Few studies have addressed the pivotal role of Tfh cells in orchestrating memory B cell and antibody-producing plasma cell responses in the context of *C. difficile* infection. A comprehensive gene expression and flowcytometric analysis of Tfh and memory B cells holds the potential to provide deep insight in the pathogenesis of rCDI, as well as for the exploration of an optimal vaccine strategy. Moreover, the impact of T cell responses, such as Th1, Th2, and Th17, remains an evolving area of research in CDI pathogenesis. Additionally, investigating the potential role of regulatory T cells in CDI susceptibility and recurrence is essential. The development of effective vaccines against *C. difficile* toxins, particularly TcdB, holds promise for preventing CDI and should be further explored. Furthermore, alternative treatments like probiotics, live biotherapeutics, and FMT offer potential ways to modulate the host immune response and microbiome to prevent CDI recurrences. Larger-scale clinical trials are needed to validate the efficacy and safety of these therapies. In the arena of antibiotic failure, a comprehensive understanding of the host immune response and leveraging the knowledge in developing alternative treatment approaches and vaccines are vital for improving CDI management and reducing its burden on public health.

## Figures and Tables

**Figure 1 tropicalmed-08-00506-f001:**
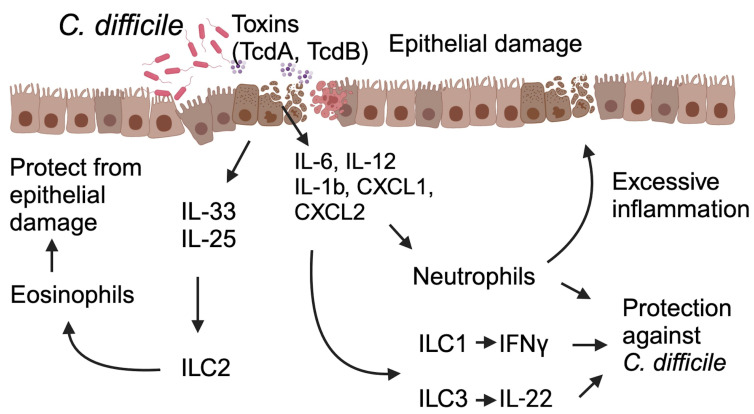
This schematic illustrates the host’s immune response during *C. difficile* infection and its impact on the infection’s outcomes. *C. difficile* toxins A and B lead to epithelial damage, triggering immune-cell activation and the release of cytokines and chemokines from both immune cells and the damaged epithelium. These signaling molecules, in turn, activate innate lymphoid cells (ILCs) and promote the recruitment of neutrophils to the site of injury. Activated ILC1 and ILC3 release Interferon IFNγ and IL-22, conferring protection against *C. difficile*. Neutrophils, though known for their protective function, may also have the potential to cause epithelial damage and have a detrimental impact on the outcome of CDI. Specific cytokines, such as IL-25 and IL-33, induce a type 2 immune response, enhancing host defense by increasing eosinophil infiltration and activation at the site of damage. Abbreviations: ILC1 (Innate Lymphoid Cell 1), ILC2 (Innate Lymphoid Cell 2), ILC3 (Innate Lymphoid Cell 3), TcdA (*C. difficile* Toxin A), TcdB (*C. difficile* Toxin B).

**Figure 2 tropicalmed-08-00506-f002:**
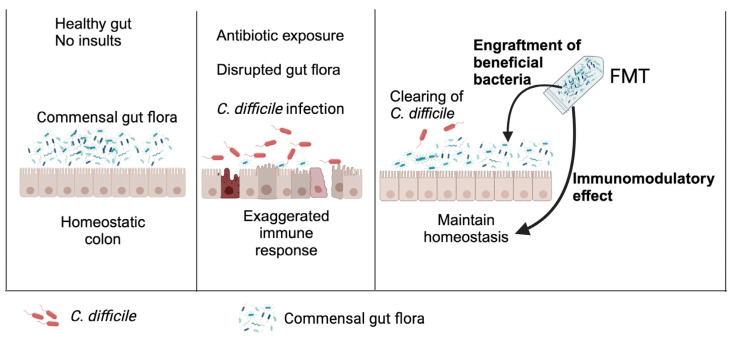
Fecal microbiota transplantation effectively clears *C. difficile* infection. Commensal gut flora, represented in green and blue, play a crucial role in protection against *C. difficile* in a healthy gut. However, the use of antibiotic therapy disrupts these commensal bacteria, rendering the gut susceptible to *C. difficile* infection (indicated as red). The resulting increased inflammation, subsequent to *C. difficile* infection, creates an environment conducive to the growth of pathobionts, further contributing to dysbiosis and negatively impacting the course of the infection. After fecal microbiota transplantation (FMT), the successful engraftment of beneficial bacteria from the FMT inoculum establishes colonization resistance against *C. difficile*. Additionally, the bacteria or bacterial-derived metabolites from the FMT inoculum exhibit anti-inflammatory or immunomodulatory effects. These effects help to restore the gut’s immune response to a homeostatic level, ultimately aiding the host in clearing the *C. difficile* infection.

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
