# Peer review of "Host Immune Responses to *Clostridioides difficile* Infection and Potential Novel Therapeutic Approaches"

_tropicalmed, 2023, doi:10.3390/tropicalmed8120506_

Round 1
Reviewer 1 Report
Comments and Suggestions for Authors
In this review, the authors focus on current discussions about the role of innate and adaptive immune responses in the prognosis of CDI, while also summarizing non-antibiome-based therapies. The whole text is logical and rich in content. However, there are still some minor problems.
1. The introduction discusses the public health significance and recurrence of CDI, and should further briefly discuss the impact of modulating the host immune response on the treatment of CDI, which easily connect with the following immune topic.
3. on page 3, Figure 1 shows "INFg", but "INFγ" in the legend.
4. Figure 2, part of bacteria were covered by words.
5. 2.1 Innate Immune Response may add a summarization, similar as the end of 2.2 Adaptive Immune Response.
Author Response
Dear Reviewer,
Thank you very much for taking the time to review this manuscript. Please see the point-by-point response to your comments below. In addition, the corresponding revisions/corrections are highlighted (in red) in the revised manuscript. Please do not hesitate to ask if you have any questions.
- The introduction discusses the public health significance and recurrence of CDI, and should further briefly discuss the impact of modulating the host immune response on the treatment of CDI, which easily connect with the following immune topic.
Author response: The authors completely agree with the reviewer. A brief discussion (with appropriate references) on the impact of modulating the host immune response on the treatment of CDI is added at the end of the introduction section. This addition will help the audience to easily connect with the following section. Thanks
- on page 3, Figure 1 shows "INFg", but "INFγ" in the legend.
Author response: The authors apologize for this inconsistent use of nomenclature. In figure 1, it is now corrected as IFNγ, which is consistent with the figure legend. Thanks
- Figure 2, part of bacteria were covered by words.
Author response: Thanks for pointing that out. In the revised manuscript, the authors corrected that in figure 2.
- 1 Innate Immune Response may add a summarization, similar as the end of 2.2 Adaptive Immune Response.
Author response: The authors agree with the reviewer. A summarization at the end of the discussion related to innate immune response is now added in the revised manuscript. Thanks
Reviewer 2 Report
Comments and Suggestions for Authors
Some words are inappropriately hyphenated and require uniform notation. (Pro-inflammatory or pro-inflammatory, biotherapeutic, etc.)
Names of bacteria should be italicized, but in many places this is not the case.
The first time an abbreviation is mentioned, the full term must be given.
In the 3.3 FMT chapter, there is a need to add explanations about Rebyota®, etc., which are currently approved FMT material by the FDA.
In the middle stage of Figure 2, it seems appropriate to present the order as antibiotic exposure – disrupted gut flora - C. difficile infection rather than the order of C. difficile infection - antibiotic exposure - disrupted gut flora.
Author Response
Dear Reviewer,
Thank you very much for taking the time to review this manuscript. Please see the point-by-point response to your comments below. In addition, the corresponding revisions/corrections are highlighted (in red) in the revised manuscript. Please do not hesitate to ask if you have any questions.
Reviewer: Some words are inappropriately hyphenated and require uniform notation. (Pro-inflammatory or pro-inflammatory, biotherapeutic, etc.)
Author response: The authors apologize for using the inappropriate use of hyphens in a few words. In the revised manuscript, these are corrected; for example, “Pro-inflammatory” is changed to “Proinflammatory”. We also have made sure that they are consistent all over the manuscript. Thanks
Reviewer: Names of bacteria should be italicized, but in many places this is not the case.
Author response: The authors apologize for this unintentional error. In the revised manuscript, we have fixed this error. Thanks
Reviewer: The first time an abbreviation is mentioned, the full term must be given.
Author response: The authors appreciate this comment. We tried to fix these in the revised manuscript. Thanks
Reviewer: In the 3.3 FMT chapter, there is a need to add explanations about Rebyota®, etc., which are currently approved FMT material by the FDA.
Author response: The authors agree entirely with the reviewer. In the revised manuscript, a brief discussion on “Rebyota” is added (with appropriate references) in the FMT sub-section. This addition will help the audience to learn the updated treatment approach in treating CDI. Thanks
Reviewer: In the middle stage of Figure 2, it seems appropriate to present the order as antibiotic exposure – disrupted gut flora - C. difficile infection rather than the order of C. difficile infection - antibiotic exposure - disrupted gut flora.
Author response: The authors agree with the reviewer. In the revised manuscript, in figure 2 (middle stage), the order is changed as antibiotic exposure – disrupted gut flora - C. difficile infection. Thanks
Reviewer 3 Report
Comments and Suggestions for Authors
This is an excellent comprehensive review on CDI and potential measures to prevent recurrence using different novel approaches. A few comments are suggested:
1. Line 49 and 53: No need to mention the names of the journals as this is uncommon to be reported. The reader can simply refer to the cited reference.
2. Line 83: They are not broad-spectrum. I suggest replacing this with something like: both have activity against other microbial flora...
3. Line 265: Please add a citation to the sentence "Fidaxomicin offers a promising alternative, with a lower C. difficile recurrence rate." Some suggested references include PMID: 26592763 and 26661400.
4. Line 265: Move the citation [70] to the end of the sentence on line 266.
5. Section 3.1 Probiotics: A very important study that is missing from this part is the large multicenter RCT by Allen, et al (PMID: 23932219). The authors found no benefit of probiotics for CDI prevention. Similar findings were also reported in a study by Heil, et al (PMID: 33972996). I think adding those arguments and controversy regarding the use of probiotics is important to indicate that the use of probiotics remains debatable/controversial. I also suggest menioning the sample sizes of these studies and the studies already mentioned in this section for the reader to draw conclusion based on the sizes of the studies.
6. Line 345: Remove the spell out "fecal microbiota transplantation" but keep the abbreviation without parentheses.
7. Line 469: You can use the abbreviation FMT here.
Author Response
Dear Reviewer,
Thank you very much for taking the time to review this manuscript. Please see the point-by-point response to your comments below. In addition, the corresponding revisions/corrections are highlighted (in red) in the revised manuscript. Please do not hesitate to ask if you have any questions.
- Line 49 and 53: No need to mention the names of the journals as this is uncommon to be reported. The reader can simply refer to the cited reference.
Author response: The authors appreciate this comment and totally agree with the reviewer. In the revised manuscript, the journal names are deleted, and readers are referred to the cited references only. Thanks
- Line 83: They are not broad-spectrum. I suggest replacing this with something like: both have activity against other microbial flora...
Author response: The authors agree with the reviewer. In the revised manuscript, we replaced this with “both demonstrate antibacterial effects against beneficial bacteria in the normal gut flora, leading to persistent dysbiosis.” Thanks
- Line 265: Please add a citation to the sentence "Fidaxomicin offers a promising alternative, with a lower C. difficile recurrence rate." Some suggested references include PMID: 26592763 and 26661400.
Author response: The authors appreciate this comment and apologize for not providing any citations before. In the revised manuscript, we added these two citations (PMID: 26592763 and 26661400) that the reviewer suggested. We believe these citations will direct the reader to the correct information. Thanks
- Line 265: Move the citation [70] to the end of the sentence on line 266.
Author response: The authors agree with the reviewer. In the revised manuscript, we moved the citation [70] to the end of the sentence on line 266. Please note that, due to the revision of the manuscript, this reference now appeared as 72, in line 283. Thanks
- Section 3.1 Probiotics: A very important study that is missing from this part is the large multicenter RCT by Allen, et al (PMID: 23932219). The authors found no benefit of probiotics for CDI prevention. Similar findings were also reported in a study by Heil, et al (PMID: 33972996). I think adding those arguments and controversy regarding the use of probiotics is important to indicate that the use of probiotics remains debatable/controversial. I also suggest menioning the sample sizes of these studies and the studies already mentioned in this section for the reader to draw conclusion based on the sizes of the studies.
Author response: The authors agree entirely with the reviewer. In the revised manuscript, in the probiotics subsection, we mentioned these two studies, which found no beneficial effect of probiotics in treating CDI. Like the reviewer, we also believe that adding those studies regarding the use of probiotics is crucial to point out that the use of probiotics remains debatable in the context of CDI. Thanks
- Line 345: Remove the spell out "fecal microbiota transplantation" but keep the abbreviation without parentheses.
Author response: In the revised manuscript, we removed the spelling of "fecal microbiota transplantation" but kept the abbreviation without parentheses. Thanks
- Line 469: You can use the abbreviation FMT here.
Author response: In the revised manuscript, we used the abbreviated FMT in that line. Thanks